# GUI-KV: Efficient GUI Agents via KV Cache with Spatio-Temporal Awareness

**Kung-Hsiang Huang**[1,3]*, **Haoyi Qiu**[2]†, **Yutong Dai**[3], **Caiming Xiong**[3], **Chien-Sheng Wu**[3]
*khshuang@amazon.com*
[1] *Amazon AGI SF Lab*
[2] *University of California, Los Angeles*
[3] *Salesforce AI Research*
https://github.com/SalesforceAIResearch/GUI-KV

**Reviewed on OpenReview:** https://openreview.net/forum?id=7262

## Abstract

Graphical user interface (GUI) agents face severe efficiency bottlenecks when processing long sequences of high-resolution screenshots, making inference costly and memory-bound. Existing KV cache compression methods, designed for natural images, remain suboptimal as they fail to exploit the unique spatial and temporal redundancies of GUIs. In this work, we first demonstrate that unlike natural images, GUI attention sparsity is uniformly high ($> 0.99$) across all transformer layers, invalidating complex layer-varying budget strategies. Building on this insight, we introduce GUI-KV, a training-free compression method that allocates a uniform budget driven by two novel mechanisms: (1) *spatial saliency guidance*, which augments attention with residual stream L2 norms to preserve semantic visual tokens; and (2) *temporal redundancy scoring*, which employs subspace projection to identify and prune historical frames that are redundant to the current view. Across six benchmarks, GUI-KV outperforms competitive baselines, often recovering near-full-cache accuracy at 10-20% budgets. Notably, on AgentNetBench, it reduces decoding FLOPs by 38.9% while increasing step accuracy by 4.1% over the full-cache baseline.

## 1 Introduction

Graphical user interface (GUI) agents, which automate tasks by interacting with graphical user interfaces, have emerged as a crucial application of vision-language models (VLMs) (Qin et al., 2025; Wang et al., 2025; Gou et al., 2025). These agents navigate complex digital environments by processing sequences of screenshots and generating actions to accomplish user-specified goals. However, the computational demands of processing high-resolution GUI screenshots with VLMs present significant efficiency challenges. For example, running inference UI-TARS-1.5-7B (Qin et al., 2025) with bfloat16 and flash attention 2 using Huggingface inference on a 5-screenshot example for a single step from OSWorld-Verified (Xie et al., 2024) can take over 15 seconds on a H200 on average.

Some work address the inference challenge by merging input visual tokens (Lin et al., 2025) or pruning visual representations at higher layers (Chen et al., 2025). However, such approaches require re-training GUI agents. A more desirable approach is key-value (KV) cache, which offers a plug-and-play solution where previously computed key and value representations from attention layers are stored in memory to avoid redundant computations during autoregressive generation. Nevertheless, caching all KV states can require extensive GPU memory. For instance, UI-TARS-1.5-7B can consume over 80GB GPU memory on a single inference when feeding 5 screenshots with a maximum steps of 50. This can easily trigger out-of-memory error for most of the consumer GPUs. Consequently, this massive memory footprint constitutes the primary bottleneck

---

*Work done at Salesforce AI Research.
†Work done during internship at Salesforce AI Research.

preventing the real-world deployment of GUI agents on local machines, edge devices, and standard consumer hardware, making efficient KV cache compression not just an optimization, but a strict prerequisite for practical adoption.

While recent work has introduced various KV cache compression techniques (Xiao et al., 2024; Zhang et al., 2023; Cai et al., 2025; Yang et al., 2024; Li et al., 2024b; Tu et al., 2025), their effectiveness on GUI agent tasks remains unexplored. GUI agent tasks present unique characteristics that distinguish them from typical vision-language understanding. Screenshots contain substantial spatial redundancy—large regions of uniform backgrounds, repeated UI elements, and static components that persist across time steps. Moreover, the sequential nature of GUI interactions introduces temporal redundancy, as consecutive screenshots often share significant visual overlap. These properties suggest that existing KV cache approaches, developed primarily for natural images and documents, may not be optimal for GUI environments.

In this work, our first contribution is the first systematic analysis for understanding if current KV cache budget allocation strategies across transformer layers remain effective for GUI agents (§3). Results show that existing approaches fail to capture the unique properties of GUI scenarios. Our analysis indicate GUI screenshots exhibit uniformly extreme and flat attention sparsity across layers ($> 0.99$), whereas natural images can have sparsity reaches as low as 0.88. Figure 1). This causes per-layer budget schedules that normalize sparsity (*e.g.*, PyramidKV (Cai et al., 2025) and VL-Cache (Tu et al., 2025)) to over-amplify tiny differences and misallocate cache. Thus, we posit that it is best to assign uniform KV cache budget to each layer for GUI agents.

Furthermore, our second contribution is **GUI-KV**, a KV cache compression method tailored for GUI agents (§4). Our approach introduces two key innovations. First, we develop a residual stream-based saliency guidance mechanism that augments attention-based scoring with the L2 norm of visual token hidden states. This design is inspired by insights from previous work showing that certain tokens act as information sinks (Darcet et al., 2024). Second, motivated by the fact that screenshots from prior steps contain overlapping information as the current frame, we introduce temporal redundancy scoring mechanism that performs QR decomposition over the current screenshots to determine redundant visual information from prior steps. To our knowledge, this is the first approach in KV cache compression that exploit inter-frame correlations. The two components work synergistically to determine the spatio-temporal saliency of each token.

Finally, our third contribution is the comprehensive experiments on two leading GUI agent models, UI-TARS-1.5-7B (Qin et al., 2025) and OpenCUA-7B (Wang et al., 2025), across six GUI agents benchmarks covering visual grounding and end-to-end evaluation in web, desktop, and mobile (§5). Our findings show that GUI-KV enables GUI agents to operate with significantly reduced memory, while maintaining high performance across six benchmarks. Empirically, GUI-KV recovers near–full-cache accuracy with modest budgets (typically 10–20%) and matches or outperforms the best competing compression baselines across tasks, occasionally even exceeding the full-cache accuracy at intermediate budgets. On efficiency, pre-filling overhead is negligible (increase $< 0.29\%$), while decoding compute drops substantially—for example, at a 40% budget with five screenshots we reduce FLOPs per decoded token by 38.9% and simultaneously improve step accuracy by +4.1%; the savings grow with more screenshots. Ablations further show complementary benefits: spatial saliency is more effective when the number of screenshots is small, temporal redundancy becomes increasingly beneficial as more screenshots are provided, and combining both yields the best performance across settings.

## 2 Related Work

**Vision Token Compression.** Recent approaches to reducing vision token computational burden fall into two categories: architectural modifications and adaptive token pruning. Chu et al. (2024) employ lightweight projector architectures with average pooling layers to compress visual tokens. Adaptive methods like LLaVA-PruMerge (Shang et al., 2024) and MADTP (Cao et al., 2024) dynamically reduce tokens based on importance scores derived from attention patterns. Chen et al. (2024) combine adaptive attention in early layers with token pruning in later stages. Recently, Chen et al. (2025) extends these ideas to GUI agents through context-aware simplification strategies.

**KV Cache Compression.** Post-training KV cache compression methods fall into four categories. Token-wise eviction strategies like StreamingLLM (Xiao et al., 2024) retain attention sinks and recent tokens for

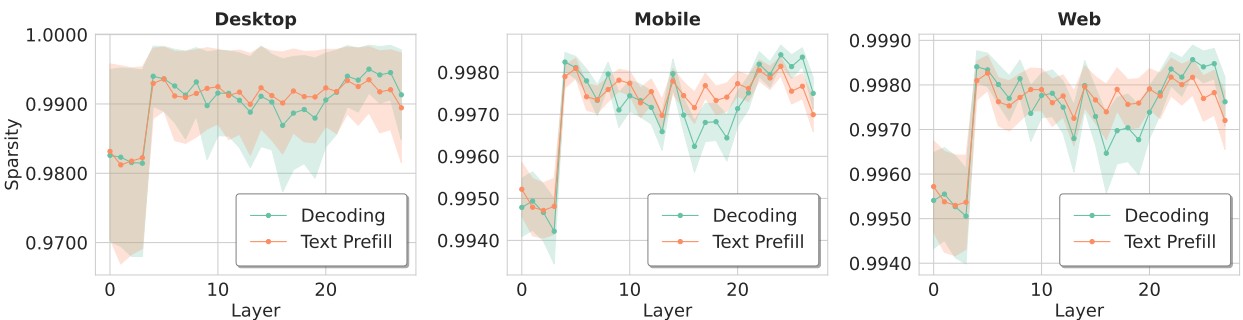

Figure 1: Attention sparsity for UI-TARS-1.5-7B across layers for screenshots from ScreenSpot-V2. All scenarios exhibit extremely high sparsity (mostly > 0.99) across all layers.

infinite-length generation, while Zhang et al. (2023) and Li et al. (2024b) identify heavy-hitter tokens, though potentially sacrificing context. Token-wise merging approaches preserve more information—Zhang et al. (2024) consolidate information from multiple tokens, and Wan et al. (2024a) employ dynamic discriminative operations based on semantic similarity. Static layer-wise reduction methods like Cai et al. (2025) apply uniform compression ratios across layers following a pyramidal structure but may overlook varying layer importance. Quantization techniques reduce memory through precision reduction, with Liu et al. (2024) demonstrating effective 2-bit asymmetric quantization and Kang et al. (2024) achieving near-lossless compression through error reduction frameworks. Most methods focus on text-based compression, overlooking multimodal challenges. While Wan et al. (2024b) address multimodal compression, they use fixed allocation strategies that ignore inter-layer attention variations. Recent works by Wan et al. (2025) and Tu et al. (2025) introduce dynamic allocation, advancing toward more adaptive multimodal compression strategies.

## 3 Preliminary & Analysis

### 3.1 GUI Agent Problem Definition

GUI agents are designed for GUI navigation tasks, where they sequentially generate actions and interact with the GUI environment to achieve a specified goal (Qin et al., 2025; Wang et al., 2025). This interaction can be formulated as a sequential decision-making process. More formally, a GUI agent task can be modeled as a Partially Observable Markov Decision Process (POMDP), defined by a tuple $(\mathcal{S}, \mathcal{A}, \mathcal{F}, \mathcal{R}, \mathcal{O})$. At each step $t$, the agent is in a state $s_t \in \mathcal{S}$, which represents the true state of the GUI environment (*e.g.*, the current desktop environment). The agent receives a partial observation $o_t \in \mathcal{O}$, which typically includes a screenshot of the GUI. The agent also has access to the natural language instruction or goal $G$. Based on the observation $o_t$ and its history of past observations and actions, the agent performs an action $a_t \in \mathcal{A}$, such as clicking or typing text. The environment then transitions to a new state $s_{t+1} \sim \mathcal{F}(s_t, a_t)$, where $\mathcal{F}$ is the state transition function. The agent receives a new observation $o_{t+1} \sim \mathcal{O}(s_{t+1})$ and a reward $r_t = \mathcal{R}(s_t, a_t)$. For interactive settings such as OSWorld-Verified, the interaction loop continues until a terminal action is generated or a maximum number of steps is reached.

### 3.2 KV Cache in Vision-Language Models

When GUI agents process sequences of observations and generate actions, the underlying VLMs must efficiently handle the multimodal inputs. A critical component in this process is the key-value (KV) cache mechanism, which avoids redundant processing. Below, we provide an overview of KV caching.

**Prefill phase.** When a VLM encodes multimodal input, the visual encoder processes the screenshot to generate visual tokens, which are then mapped to a unified embedding space through a projection layer. Concurrently, any language input, such as the task goal, is tokenized and embedded. The language model component of VLM processes these tokens in parallel. During this process, the key and value states are cached in GPU memory to avoid recomputation in subsequent steps.

**Decoding phase.** After the prefill phase, the model generates action tokens auto-regressively. Each step produces new key-value pairs corresponding to the newly generated token, which are appended to the cache.

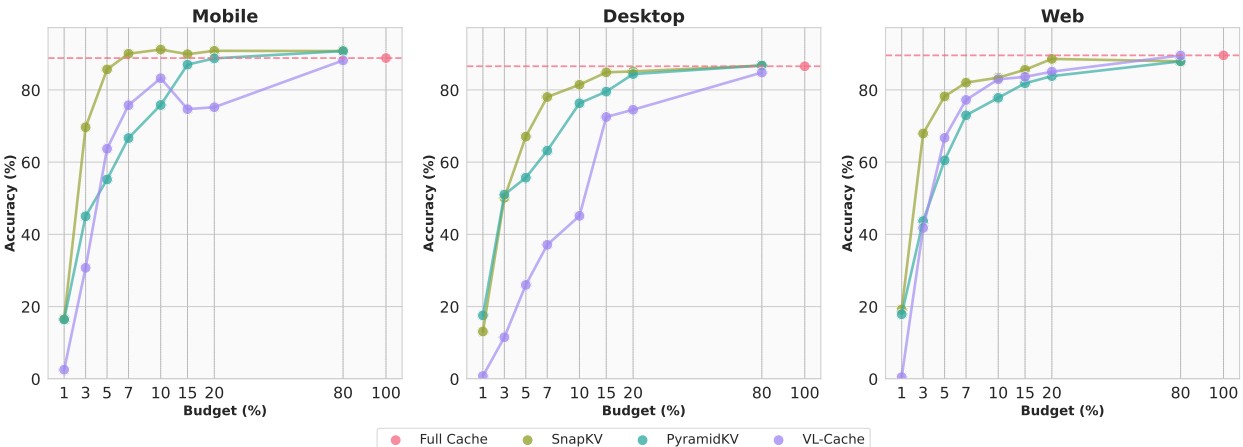

Figure 2: Performance comparison of KV cache methods on ScreenSpotV2. SnapKV's uniform allocation outperforms approaches that allocate varying budgets.

For GUI agents that maintain history containing multiple high-resolution screenshots and long action outputs, the KV cache can grow substantially, creating memory bottlenecks that limit the agent's ability to process long interaction sequences. To address this memory bottleneck, researchers have developed KV cache compression techniques that maintain only a subset of the full cache while minimizing accuracy loss. These algorithms operate along two main dimensions: (1) budget allocation across layers, and (2) token selection policies.

**Budget Allocation.** Given the transformer's layered architecture, early work allocates equal KV cache slots to each layer (Xiao et al., 2024; Zhang et al., 2023), while recent work has shown that different layers have varying sensitivity to cache reduction (Cai et al., 2025; Yang et al., 2024).

**Token Scoring Policy.** For a given cache budget $\gamma \in (0, 1]$ (*i.e.*, keeping $\gamma$ of KV states for a layer), we need a scoring function $\psi$ to rank token importance. In practice, this scoring is typically computed using *observation tokens*, the last $\omega$ input tokens that serve as queries to determine which KV states are most relevant to retain (Li et al., 2024b). Let $n$ be the number of total input prompt tokens, with indices $\mathcal{C} = \{1, 2, \ldots, n\}$. The scoring function $\psi : \mathcal{C} \to \mathbb{R}^n$ assigns importance scores to each token based on the attention they receive from these observation tokens. Based on the scoring function $\psi$ and budget $\gamma$, we select the indices with top scores:

$$\mathcal{C}_\psi = \{i \in \mathcal{C} : |\{j \in \mathcal{C} : \psi(i) > \psi(j)\}| < \lceil \gamma \cdot n \rceil\} \tag{1}$$

## 3.3 High Spatial Redundancy in GUI

GUI screenshots exhibit fundamentally different visual characteristics compared to natural images, particularly in terms of spatial redundancy. Unlike natural scenes that contain rich textures and continuous variations, GUI interfaces are dominated by large uniform regions, such as blank spaces, solid-colored backgrounds, and repeated UI elements. This structural difference has profound implications for attention patterns in VLMs and, consequently, for KV cache compression strategies.

To investigate whether current KV cache budget allocation strategies remain effective for GUI agents, we analyze attention sparsity patterns across different GUI environments. We measure the sparsity of the attention score matrix in different transformer layers during the prefill and decoding phases, where the attention scores are computed from observation tokens (the most recent decoder tokens) to all cached tokens. First, for each head $h$, we let $\tilde{\boldsymbol{Q}}^h \in \mathbb{R}^{\omega \times d_h}$ denote the queries of the last $\omega$ input tokens, $\boldsymbol{K}^h \in \mathbb{R}^{(n-\omega) \times d_h}$ be the previous key states, and $\boldsymbol{A}^h = \text{softmax}(\frac{\tilde{\boldsymbol{Q}}^h \boldsymbol{K}^{h\top}}{\sqrt{d_h}}) \in \mathbb{R}^{\omega \times (n-\omega)}$ be the subset of attention scores from the last $\omega$ tokens to the previous tokens. Then, we apply a filter with a relative threshold $p$ to the attention score matrix $\boldsymbol{A}^h$:

$$\text{ThresholdFilter}(\boldsymbol{A}^h, p)_{ij} = \begin{cases} \boldsymbol{A}_{ij}^h & \text{if } \boldsymbol{A}_{ij}^h \geq p \cdot \max_j(\boldsymbol{A}_{ij}^h) \\ 0 & \text{otherwise} \end{cases} \tag{2}$$

where threshold $p \in (0, 1)$ controls the strength of induced sparsification. Following Tu et al. (2025), we heuristically set $p = 1\%$, such that the filtered-out scores have an insignificant impact on the output of the

transformer layer. After filtration, we calculate sparsity $\varsigma \in [0, 1]$ as:

$$\varsigma := \frac{\sum_{i+n-\omega \geq j} \mathbb{1}[\text{ThresholdFilter}(\boldsymbol{A}^h, p)_{ij} = 0]}{|\{\boldsymbol{A}^h_{ij} : i + n - \omega \geq j\}|} \tag{3}$$

which represents the percentage of zero entries. This metric captures how concentrated the attention distribution is—higher sparsity indicates that the model focuses on fewer tokens.

We compute $\varsigma$ across transformer layers using GUI agent trajectories from ScreenSpotV2 (Wu et al., 2025), which provides diverse interaction scenarios spanning web, mobile, and desktop environments. Figure 1 reveals a striking finding: *attention sparsity in GUI screenshots consistently averages above 0.99 across all transformer layers, with the lowest values around 0.98*. These are significantly higher than natural images, which average 0.92 with minimums around 0.88 (Tu et al., 2025). This extreme sparsity remains remarkably stable across layers, forming an almost flat line when plotted against layer depth. In contrast, natural images exhibit decreasing sparsity in deeper layers, reflecting the model's need to integrate increasingly diverse visual features for complex scene understanding.

This uniformly high sparsity suggests that existing budget allocation strategies may be suboptimal for GUI agents. Methods like PyramidKV (Cai et al., 2025) and VL-Cache (Tu et al., 2025) allocate cache budgets based on the assumption that attention sparsity varies significantly across layers. However, when sparsity differences are minimal, their normalization procedures can artificially create budget variations that do not reflect actual attention patterns. Specifically, VL-Cache normalizes attention sparsity scores before computing per-layer budgets, which amplifies small numerical differences into substantial budget disparities even when the underlying attention patterns are nearly identical.

To validate this hypothesis, we compare three budget allocation strategies on ScreenSpotV2 tasks: (1) Li et al. (2024b), which allocate equal budget across all layers; (2) PyramidKV, which heuristically assigns more budget to shallower layers; and (3) VL-Cache, which computationally determines budgets based on normalized attention sparsity.

By evaluating a strict uniform budget against heuristic and proxy-driven layer-wise allocations, we effectively conduct a macro-level ablation of layer-wise eviction sensitivity. As shown in Figure 2, SnapKV achieves significantly better performance than both PyramidKV and VL-Cache across various compression ratios. **This empirical gap demonstrates that layer-wise eviction sensitivity in GUI agents is flat; forcing a uniform budget prevents the catastrophic eviction of critical tokens that occurs when non-uniform methods incorrectly penalize specific layers.**

The performance gap widens at higher compression ratios, particularly when retaining only 5% to 20% of the original cache budget, suggesting that misallocated budgets become increasingly detrimental when cache resources are scarce. These findings demonstrate that **prior studies on budget allocation strategies, developed primarily for natural images and text, are not transferable to GUI agent tasks**.

Since uniform allocation is a special case of dynamic allocation, one might assume dynamic methods simply need better tuning for GUI agents. However, our sparsity analysis reveals that for GUI agents, the optimal dynamic allocation mathematically collapses to a uniform distribution. Because there is no meaningful layer-wise variance to exploit, any dynamic algorithm attempting to reallocate budgets based on these flat sparsity scores will inevitably fit to numerical noise rather than genuine architectural needs.

## 4 GUI-KV

In this section, we introduce our method GUI-KV, which enhances KV cache compression for GUI agents through two novel mechanisms: (1) spatial saliency guidance that leverages the L2 norm of visual token hidden states to better identify important tokens (§4.1), and (2) temporal redundancy-aware scoring that exploits the sequential nature of GUI interactions (§4.2). An overview of GUI-KV is shown in Figure 3.

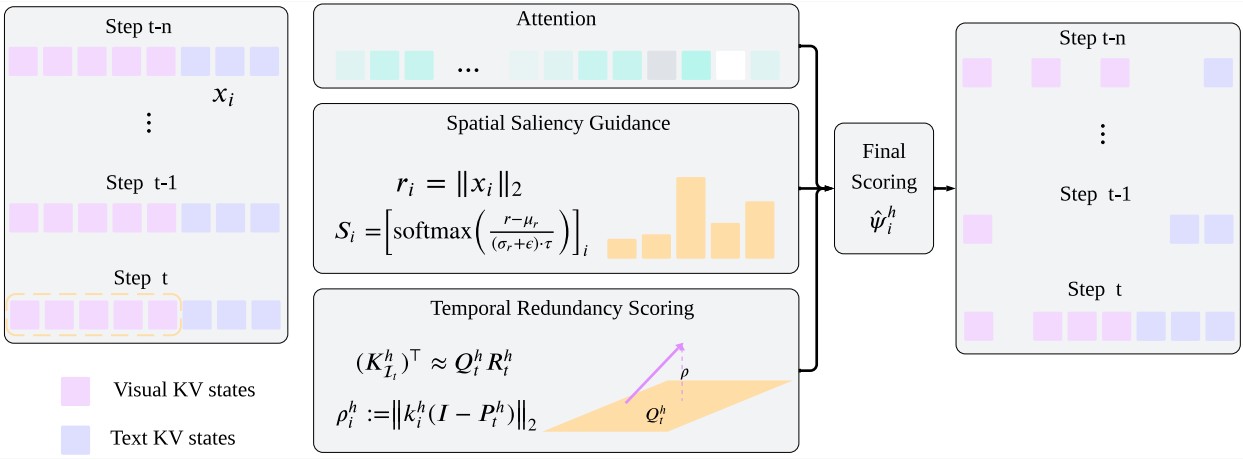

Figure 3: Overview of GUI-KV. **Spatial Saliency Guidance** utilizes the L2 norms ($r_i$) of Step $t$ visual representations to identify intrinsically important features. **Temporal Redundancy Scoring** constructs a subspace ($Q_t^h$) from Step $t$ keys to prune historical tokens that possess low projection residuals ($\rho$), indicating redundancy. The final score combines these signals to retain only spatially salient and temporally unique KV states.

## 4.1 Spatial Saliency Guidance

GUI agents process screenshots, where only a small fraction of the visual tokens are typically relevant to the task at hand. While attention mechanisms are designed to identify these salient regions, they are not always sufficient. Furthermore, studies on vision transformers suggest that not all visual tokens contribute equally to the model's understanding. Some tokens act as "register-like" information sinks, aggregating content from across the visual input, while others are less informative (Darcet et al., 2024). This motivates us to supplement attention-based selection with a more direct measure of a token's content strength, allowing us to better identify salient visual information.

Let $\mathcal{I}_t$ and $\mathcal{T}_t$ denote the set of visual token indices and text token indices in the current step (step $t$), and let $x_i \in \mathbb{R}^d$ be the residual-stream hidden state of token $i$ at a given layer. We define per-token magnitudes with $r_i = \|x_i\|_2$ and $\mu_r$ with a sample mean of $\sigma_r$ and standard deviation of $\boldsymbol{r}$. Saliency score can be computed by applying softmax to the standardized norms:

$$S_i = \left[ \text{softmax}\left( \frac{\boldsymbol{r} - \mu_r}{(\sigma_r + \epsilon) \cdot \tau} \right) \right]_i, \tag{4}$$

where $\boldsymbol{r} = [r_i]_{i \in \mathcal{I}_t}$, $\tau > 0$ is a temperature hyperparameter, and $\epsilon = 10^{-8}$ for numerical stability. Intuitively, if attention is "demand," then $S_i$ is the normalized "payload." The raw norm $r_i$ remains a reliable proxy for the strength of the information carried by token $i$ because the learned value and output projections are approximately norm-preserving (Xiong et al., 2020). Computing $r_i$ on the residual-stream hidden state before self-attention ensures that $x_i$ captures the accumulated content up to the given layer. The subsequent standardization and softmax preserve the ordering induced by $r_i$ while yielding saliency weights that are comparable across samples. By combining this normalized measure of absolute content strength with the relational signal in the attention scores $\boldsymbol{A}_{j,i}$, our method maintains both content magnitude and contextual relevance in token selection.

We therefore score the current visual tokens by a linear combination of attention and norm:

$$\psi_i^h = \begin{cases} A_i^h + \alpha S_i & \text{if } i \in \mathcal{I}_t \text{ (visual tokens)} \\ A_i^h & \text{if } i \in \mathcal{T}_t \text{ (text tokens)} \end{cases} \tag{5}$$

where $\alpha > 0$ balances the two signals. We prefer an additive form over a product $A_i \cdot S_i$ for numerical stability and for easier calibration of $\alpha$. §B.3 discusses the various $S_i$ we experimented.

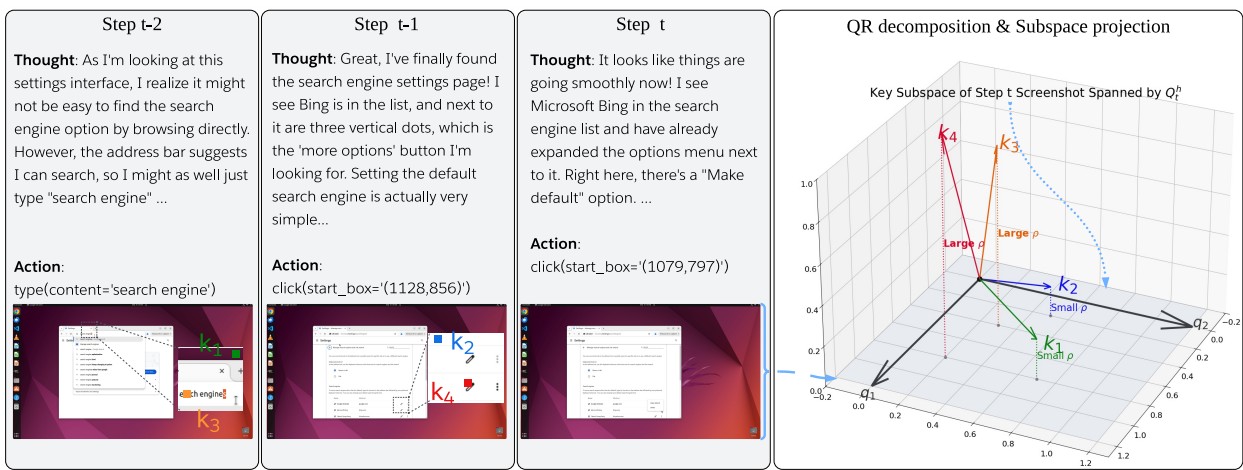

Figure 4: Illustration of our temporal redundancy scoring mechanism. We first perform QR decomposition on the current screenshot at step $t$ to obtain a subspace spanned by $\boldsymbol{Q}_t^h$. Then, we project key vectors $k_i's$ of visual tokens from previous frames onto $\boldsymbol{Q}_t^h$. The magnitude of the residual $\rho$ (the component orthogonal to the subspace) quantifies the visual token's non-redundancy. In this example, $k_3$ and $k_4$ are less redundant than $k_1$ and $k_2$ since they have larger residuals orthogonal to $\boldsymbol{Q}_t^h$. This serves as a qualitative visualization of our pruning criteria: static background elements ($k_1$, $k_2$) are pruned, while semantically critical UI changes ($k_3$, $k_4$) are selected and preserved.

## 4.2 Temporal Redundancy Scoring

Consecutive GUI screenshots share substantial structure (*e.g.*, static backgrounds, persistent UI elements), while only a small subset of regions changes between steps. We capitalize on this temporal redundancy by preferentially retaining tokens from earlier frames whose content is not already represented by the most recent frame. At step $t$, let $\mathcal{I}_{t-1}, \dots, \mathcal{I}_{t-k}$ denote the token indices of $k$ previous images. For a head $h$ of dimensionality $d_h$, let $\boldsymbol{K}^h \in \mathbb{R}^{T \times d_h}$ be the key matrix at the selection layer for all tokens in the prompt. Let $n_t = |\mathcal{I}_t|$ denote the number of tokens in the last image, and let $\boldsymbol{K}_{\mathcal{I}_t}^h$ be the submatrix of $\boldsymbol{K}^h$ containing the rows for indices in $\mathcal{I}_t$. We compute a rank-$r$ QR decomposition:

$$(\boldsymbol{K}_{\mathcal{I}_t}^h)^\top \approx \boldsymbol{Q}_t^h \boldsymbol{R}_t^h, \qquad \boldsymbol{Q}_t^h \in \mathbb{R}^{d_h \times r}, \tag{6}$$

where $r$ is a fixed rank and $\boldsymbol{Q}_t^h$ has orthonormal columns spanning a low-dimensional subspace of the last image's key space. We define the projector onto this subspace as $\boldsymbol{P}_t^h := \boldsymbol{Q}_t^h \boldsymbol{Q}_t^{h\top} \in \mathbb{R}^{d_h \times d_h}$. For any previous-image token $i \in \mathcal{I}_{t-\ell}$, let $k_i^h \in \mathbb{R}^{1 \times d_h}$ denote the $i$-th row of $\boldsymbol{K}^h$. We define its redundancy score relative to the last-image subspace as

$$\rho_i^h := \left\| k_i^h (\boldsymbol{I} - \boldsymbol{P}_t^h) \right\|_2, \tag{7}$$

so that tokens whose information lies largely in the span of the last image (small $\rho_i^h$) are deemed redundant, while large $\rho_i^h$ indicates genuinely new directions worth keeping. Computationally, for each head we form $\boldsymbol{Q}_t^h$ once from the last image and score previous-image tokens with two matrix multiplications per image, $\boldsymbol{K}_{\mathcal{I}_j}^h \mapsto (\boldsymbol{K}_{\mathcal{I}_j}^h \boldsymbol{Q}_t^h) \boldsymbol{Q}_t^{h\top}$ for $j < t$. This mechanism is particularly effective for consecutive GUI states with minor UI changes; static background tokens will yield near-zero residuals and be seamlessly pruned, while tokens capturing the localized UI changes maintain high residuals and are preserved, ensuring the agent retains the semantic history of its actions. Figure 4 illustrates an overview of our method.

## 4.3 Spatial and Temporal Combination

To combine spatial saliency with temporal redundancy, we compute $\rho_i^h$ for all tokens in $\mathcal{I}_{t-1} \cup \dots \cup \mathcal{I}_{t-k}$ and determine a percentile threshold per head. $\gamma$, we set the threshold as

$$\tilde{\rho}^h = \text{Percentile}_{1-\gamma}\big(\{\rho_j^h : j \in \mathcal{I}_{t-1} \cup \dots \cup \mathcal{I}_{t-k}\}\big). \tag{8}$$

Crucially, as shown in the above equation, text tokens and current-frame visual tokens bypass the temporal redundancy filter entirely. This guarantees that explicit textual instructions and the agent's immediate visual context are never erroneously penalized by inter-frame redundancy checks. We then define the final scoring function that combines both spatial saliency and temporal redundancy:

$$\hat{\psi}_i^h = \begin{cases} A_i^h + \alpha S_i & \text{if } i \in \mathcal{I}_t \text{ (current frame visual tokens)} \\ A_i^h \cdot \mathbb{1}[\rho_i^h \geq \tilde{\rho}^h] & \text{if } i \in \mathcal{I}_{t-1} \cup \cdots \cup \mathcal{I}_{t-k} \text{ (previous frames visual tokens)} \\ A_i^h & \text{if } i \in \mathcal{T} \text{ (text tokens)} \end{cases} \tag{9}$$

For each head $h$, we select the top-scoring tokens:

$$\mathcal{C}_{\hat{\psi}^h}^h = \{i \in \mathcal{C} : |\{j \in \mathcal{C} : \hat{\psi}_i^h > \hat{\psi}_j^h\}| < \lceil n \cdot \gamma \rceil\}, \tag{10}$$

where $\gamma \in (0, 1]$ is the budget applied uniformly across all heads.

The two mechanisms work synergistically: spatial saliency guidance identifies important visual tokens based on their attention reception and representational strength, while temporal redundancy scoring filters out previous-frame tokens that duplicate information already present in the current frame. This combination ensures we retain tokens that are both intrinsically important and temporally unique, maximizing the information content of the compressed KV cache. We provide a detailed algorithmic description of our method in Algorithm 1 and hyper-parameter selection in §C.

## 5 Experiments

### 5.1 Experimental Settings

**Benchmarks.** We assess the effectiveness of GUI-KV on six benchmarks. For *GUI visual grounding*, which evaluates the ability to locate specific UI elements from natural language descriptions, we use ScreenSpotV2 (Wu et al., 2025) and ScreenSpot-Pro (Li et al., 2025). For *offline agent evaluation*, which tests action prediction on pre-collected interaction trajectories, we employ AndroidControl (Li et al., 2024a), Multimodal-Mind2Web (Deng et al., 2023), and AgentNetBench (Wang et al., 2025). For *online agent evaluation*, which measures real-time task completion in live environments and inherently tests long-horizon temporal stability over multi-step interactions, we evaluate on OSWorld-Verified (Xie et al., 2024).

**Metrics.** We employ different evaluation metrics tailored to each benchmark. For ScreenSpot-V2 and ScreenSpot-Pro, we measure *click accuracy*, defined as the proportion of test samples where the model's predicted coordinate falls within the ground truth bounding box. For OSWorld-Verified, we use *success rate*, determined by whether a series of operations are successfully executed and specific milestones are achieved. For the remaining benchmarks, we adopt *step accuracy*, which assesses whether a single predicted step contains the correct operation (*e.g.*, click, write) and arguments (*e.g.*, coordinate or textual content). We follow the same evaluation settings as UGround (Gou et al., 2025).

**Models.** We evaluate two top-performing GUI agent models, UI-TARS-1.5-7B (Qin et al., 2025) and OpenCUA-7B (Wang et al., 2025). These two models share the same vision encoder but employ distinct language models, position encoding strategies, and training methods. UI-TARS-1.5-7B uses Qwen2.5 as the language model and was trained with supervised fine-tuning (SFT) and Direct Preference Optimization (DPO) (Rafailov et al., 2023), where as OpenCUA-7B's language model is based on Qwen2 and trained with SFT only. Because UI-TARS-1.5-7B and OpenCUA-7B rely on different language models and training paradigms, they serve as robust, independent backbones to validate the generalizability of our compression method. Additionally, greedy decoding is used across all settings.

**Baselines.** We consider three competitive and representative baselines: (1) SnapKV (Li et al., 2024b) is a classic and effective method that assigns a fixed budgets across all layers (2) PyramidKV (Cai et al., 2025) assigns a pyramid-shaped budgets to each layer based on herustics (3) VL-Cache (Tu et al., 2025) determines the budget for each layer by computing the corresponding attention sparsity.

Table 1: Performance comparison of different KV cache compression methods across GUI agent benchmarks. Δ represents the average absolute improvement of our method (GUI-KV) over the best performing baseline across all budget levels, excluding the 100% (full cache) budget.

| Dataset | Model | Method | Budget | | | | | | | | | Δ |
|---|---|---|---|---|---|---|---|---|---|---|---|---|
| | | | 1% | 3% | 5% | 10% | 15% | 20% | 40% | 80% | 100% | |
| ScreenSpotPro | UI-TARS-1.5-7B | SnapKV | **7.0** | 23.5 | 30.0 | 32.4 | 35.7 | 37.8 | 40.8 | **42.3** | 42.6 | |
| | | PyramidKV | 4.6 | 13.9 | 20.8 | 26.4 | 29.5 | 33.5 | 38.8 | **42.3** | 42.6 | |
| | | VL-Cache | 0.1 | 10.3 | 15.8 | 18.7 | 20.3 | 23.3 | 25.3 | 26.8 | 42.6 | |
| | | GUI-KV | 6.8 | **24.0** | **30.5** | **33.3** | **36.1** | **38.9** | **41.2** | 42.1 | 42.6 | +0.4 |
| | OpenCUA-7B | SnapKV | 10.1 | **16.9** | 18.2 | 17.0 | 15.3 | 17.1 | 14.7 | 27.1 | 44.6 | |
| | | PyramidKV | **15.4** | 14.2 | 14.5 | 11.4 | 14.5 | 12.5 | 13.3 | 14.5 | 44.6 | |
| | | VL-Cache | 1.6 | 13.8 | 15.1 | 15.2 | 12.8 | 16.3 | 15.2 | 14.5 | 44.6 | |
| | | GUI-KV | 10.1 | 16.7 | **19.4** | **18.5** | **17.3** | **19.3** | **18.3** | **28.0** | 44.6 | +1.4 |
| ScreenSpotV2 | UI-TARS-1.5-7B | SnapKV | 16.3 | 62.6 | 77.0 | 83.4 | 85.3 | 86.8 | **88.2** | 88.5 | 88.3 | |
| | | PyramidKV | **17.3** | 46.6 | 57.1 | 67.6 | 76.6 | 82.8 | 85.8 | 88.5 | 88.3 | |
| | | VL-Cache | 1.3 | 28.0 | 52.1 | 63.4 | 70.4 | 76.9 | 88.1 | 87.5 | 88.3 | |
| | | GUI-KV | 16.7 | **63.6** | **79.1** | **83.5** | **85.7** | **87.9** | **88.2** | **89.3** | 88.3 | +0.7 |
| | OpenCUA-7B | SnapKV | **41.9** | 70.7 | **73.4** | 73.4 | 73.7 | 63.6 | 67.3 | 74.6 | 92.7 | |
| | | PyramidKV | 16.1 | 34.6 | 41.1 | 41.0 | 41.0 | 63.0 | 57.2 | 74.4 | 92.7 | |
| | | VL-Cache | 1.1 | 50.3 | 57.2 | 60.4 | 67.8 | 55.5 | 65.0 | **76.2** | 92.7 | |
| | | GUI-KV | **41.9** | **72.1** | 72.8 | **75.1** | 74.5 | 67.9 | 69.5 | 75.9 | 92.7 | +1.4 |
| AndroidControl | UI-TARS-1.5-7B | SnapKV | **18.5** | **26.0** | 28.9 | 39.0 | 41.9 | 44.5 | **47.4** | 46.5 | 49.9 | |
| | | PyramidKV | 11.1 | 20.7 | 24.7 | 28.3 | 31.4 | 36.8 | 44.3 | 46.5 | 49.9 | |
| | | VL-Cache | 2.1 | 6.0 | 8.6 | 24.5 | 39.5 | 42.7 | 46.1 | 45.9 | 49.9 | |
| | | GUI-KV | 18.4 | 25.7 | **30.1** | **41.2** | **44.7** | **46.0** | 45.9 | **46.8** | 49.9 | +0.8 |
| | OpenCUA-7B | SnapKV | **3.1** | 15.1 | 18.5 | 21.3 | 27.6 | 23.7 | 28.6 | 23.9 | 41.1 | |
| | | PyramidKV | 1.9 | 11.1 | 15.1 | 21.3 | 27.3 | **27.4** | 21.8 | 24.5 | 41.1 | |
| | | VL-Cache | 0.0 | 0.2 | 2.5 | 11.4 | 12.7 | 19.9 | 27.5 | 3.5 | 41.1 | |
| | | GUI-KV | **3.1** | **15.9** | **20.1** | **23.8** | **30.2** | 23.3 | **30.9** | **31.7** | 41.1 | +2.2 |
| Multimodal-Mind2Web | UI-TARS-1.5-7B | SnapKV | 0.5 | **3.6** | 5.7 | 9.8 | 14.1 | 16.4 | 20.5 | 22.7 | 23.2 | |
| | | PyramidKV | **0.9** | 2.3 | 3.7 | 5.7 | 9.0 | 10.1 | 19.1 | 22.7 | 23.2 | |
| | | VL-Cache | 0.0 | 0.1 | 0.0 | 0.4 | 3.3 | 5.4 | 19.7 | **22.8** | 23.2 | |
| | | GUI-KV | 0.5 | 3.4 | **6.5** | **12.7** | **16.5** | **18.4** | **21.9** | 22.8 | 23.2 | +1.2 |
| | OpenCUA-7B | SnapKV | **0.5** | 2.8 | **5.8** | **10.0** | 12.6 | **14.2** | **16.6** | 0.1 | 34.2 | |
| | | PyramidKV | 0.0 | 0.6 | 1.7 | 5.3 | 6.5 | 9.5 | 0.0 | 0.0 | 34.2 | |
| | | VL-Cache | 0.0 | 0.0 | 0.0 | 0.0 | 0.1 | 0.0 | 13.2 | 0.3 | 34.2 | |
| | | GUI-KV | 0.4 | **3.1** | 4.4 | 9.4 | **13.7** | **14.2** | 15.6 | **13.1** | 34.2 | +1.4 |
| AgentNetBench | UI-TARS-1.5-7B | SnapKV | 0.3 | 0.7 | 1.6 | 3.0 | 8.1 | 10.6 | 19.0 | 19.0 | 17.5 | |
| | | PyramidKV | **0.5** | 0.4 | 0.8 | 1.5 | 2.4 | 4.4 | 10.0 | 19.0 | 17.5 | |
| | | VL-Cache | **0.5** | **2.6** | **1.9** | 5.7 | 2.2 | 2.2 | 2.9 | 2.3 | 17.5 | |
| | | GUI-KV | 0.2 | 1.0 | 1.8 | **6.1** | **11.9** | **16.3** | **21.6** | **22.6** | 17.5 | +2.3 |
| | OpenCUA-7B | SnapKV | 1.2 | 4.5 | 4.6 | 7.7 | 11.2 | 10.1 | 11.1 | 28.3 | 66.8 | |
| | | PyramidKV | 0.2 | 0.9 | 1.1 | 2.8 | 4.5 | 6.7 | 17.9 | **29.0** | 66.8 | |
| | | VL-Cache | 0.0 | 0.6 | 0.7 | 1.8 | 0.5 | 4.5 | 0.4 | 0.0 | 66.8 | |
| | | GUI-KV | **1.6** | **7.6** | **8.2** | **12.6** | **14.1** | **16.7** | **18.7** | 27.6 | 66.8 | +3.6 |
| OSWorld-Verified | UI-TARS-1.5-7B | SnapKV | 2.2 | 3.2 | 6.9 | **16.1** | 16.6 | 18.4 | 19.3 | 22.7 | 26.0 | |
| | | PyramidKV | 1.9 | 2.8 | 1.9 | 6.3 | 9.5 | 15.0 | 15.9 | 20.0 | 26.0 | |
| | | VL-Cache | 0.5 | 0.0 | 0.0 | 0.0 | 0.0 | 16.7 | 15.2 | 19.8 | 26.0 | |
| | | GUI-KV | **2.5** | **3.4** | **8.9** | **16.1** | **18.0** | **20.7** | **25.1** | 22.8 | 26.0 | +1.5 |
| | OpenCUA-7B | SnapKV | 0.5 | 0.5 | 0.5 | **1.1** | **1.3** | 1.3 | 1.3 | 16.8 | 21.4 | |
| | | PyramidKV | 0.5 | 0.5 | 0.5 | 0.5 | 0.8 | **1.6** | **1.7** | 13.4 | 21.4 | |
| | | VL-Cache | 0.5 | 0.5 | 0.5 | 1.0 | **1.3** | 1.4 | 1.3 | 15.4 | 21.4 | |
| | | GUI-KV | 0.5 | **0.8** | **0.6** | 0.8 | 1.0 | **1.6** | **1.7** | **17.5** | 21.4 | +0.2 |

## 5.2 Accuracy Evaluation

Table 1 displays the comparison between GUI-KV and baseline methods on the evaluated six benchmarks. We have the following observations:

**GUI-KV is the most effective in retaining Full KV cache performance.** Across six benchmarks and budgets from 1% to 80%, GUI-KV consistently matches or surpasses the best competing compression

Table 2: Efficiency analysis conducted on AgentNetBench, assessed by MFLOPs per decoded token.

| # Screenshots | KV Cache | $\gamma$ (%) | MFLOPs/ decoded token |
|---|---|---|---|
| 3 | Full Cache | 100 | 213.2 |
| | GUI-KV | 20 | 123.6 (-42.0%) |
| | GUI-KV | 40 | 145.3 (-31.8%) |
| 5 | Full Cache | 100 | 290.4 |
| | GUI-KV | 20 | 139.5 (-52.0%) |
| | GUI-KV | 40 | 177.4 (-38.9%) |
| 10 | Full Cache | 100 | 471.5 |
| | GUI-KV | 20 | 175.3 (-62.8%) |
| | GUI-KV | 40 | 249.9 (-47.0%) |

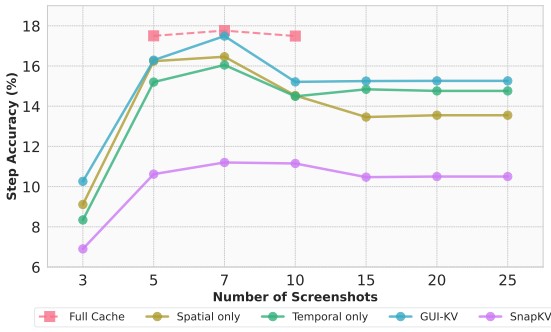

Figure 5: Ablation studies on AgentNetBench with various number of screenshots.

schemes, recovering near – full-cache accuracy with modest budgets (often 10–20% for UI-TARS-1.5-7B). On ScreenSpotPro and ScreenSpotV2, 10–20% already closes most of the gap, and by 40% the gap is negligible. On four of six benchmarks, GUI-KV achieves positive average absolute gains over the best baseline at sub-100% budgets, ranging from +0.7 to +3.6, with the largest improvements on *online and offline agent evaluation* tasks (*e.g.*, +2.2 on AndroidControl and +3.6 on AgentNetBench for OpenCUA-7B). Notably, on multiple datasets such as ScreenSpotV2 and AgentNetBench, GUI-KV slightly exceeds the full-cache accuracy at 40-80% budgets, suggesting that targeted KV pruning can *reduce long-context distraction.* By systematically pruning highly redundant background tokens from past frames, GUI-KV effectively acts as a filter, preventing the model from being distracted by irrelevant historical context. Compared to baselines, GUI-KV exhibits smoother, near-monotonic gains as the budget increases and avoids the high-budget regressions that several baselines show; at ultra-low budgets (1–3%), a baseline may edge out GUI-KV on isolated cells, but GUI-KV reliably overtakes by 5–10% and then dominates across mid and high budgets.

**UI-TARS-1.5-7B is more robust than OpenCUA-7B against evicting KV states.** UI-TARS-1.5-7B retains a higher fraction of full-cache performance under aggressive compression, reaching near saturation by 10–20% budget on visual grounding (*e.g.*, 83–89% on ScreenSpotV2) and maintaining strong step accuracy on action prediction (*e.g.*, 41.2% at 10% and 46.0% at 20% on AndroidControl with GUI-KV. In contrast, OpenCUA-7B is more sensitive to KV eviction, showing larger drops and non-monotonic behavior across budgets for multiple methods (*e.g.*, pronounced fluctuations on ScreenSpotPro and Multimodal-Mind2Web). Even with GUI-KV, OpenCUA-7B typically requires more budget to close the gap to full-cache performance and still trails UI-TARS-1.5-7B at comparable budgets on the harder action-prediction tasks (*e.g.*, 23.8% at 10% and 23.3% at 20% on AndroidControl). These trends indicate that UI-TARS-1.5-7B's architecture and training methods are intrinsically more tolerant to KV state pruning, while OpenCUA-7B benefits disproportionately from GUI-KV but remains less robust overall. Crucially, the fact that GUI-KV consistently delivers state-of-the-art compression on both a highly robust architectures demonstrates that our spatio-temporal pruning strategy generalizes effectively across diverse underlying architectures and training pipelines.

## 5.3 Efficiency Analysis

We quantify efficiency by reporting MFLOPs per token for pre-filling and decoding on UI-TARS-1.5-7B evaluated on AgentNetBench. We vary the budget $\gamma$ as well as the number of screenshots (including current and the most recent previous frames). As shown in Table 2, when $\gamma = 40\%$ in the 5 screenshot settings, we observe a 38.9% reduction in MFLOPs per decoded tokens and an absolute increase in step accuracy by 4.1% compared to the full cache baseline (see Table 1). The savings grow with more screenshots, indicating larger gains under image-heavy contexts. Importantly, because auto-regressive decoding is heavily memory-bandwidth bound, physically evicting these KV states linearly reduces the memory I/O required per token. Thus, our reported MFLOP reductions directly correlate with alleviated memory bandwidth pressure. Pre-filling overhead is negligible (increase $< 0.29\%$ across all settings) as the QR step operates on the $(d_h \times n_t)$ block for the last image and costs $O(d_h^2 n_t)$ per head, where $d_h$ is the head dimension and $n_t = |\mathcal{I}_t|$ is the number of visual tokens in the current image. This is dominated by the $O(n^2 d_h)$ attention cost per head during pre-filling with total sequence length $n$, so the added computation is negligible, while decoding compute drops substantially. As detailed in Table 3, the implementation overhead of our method, including QR decomposition and memory gathering, accounts for less than 2% of the compute cost, ensuring

that theoretical FLOP savings translate to practical decoding speedups. Overall, GUI-KV achieves significant decoding FLOP reductions with minimal pre-filling overhead.

## 5.4 Ablation Studies

We analyze the contribution of the two components of GUI-KV—spatial saliency guidance (§4.1) and temporal redundancy scoring (§4.2)—on AgentNetBench using UI-TARS-1.5-7B with a budget of $\gamma = 20\%$ while varying the number of screenshots. As shown in Figure 5, among the variants, the spatial-only guidance is more effective with fewer screenshots ($\leq 7$), while the temporal-only guidance excels as more screenshots are added, capitalizing on cross-frame redundancy. Combining both guidance, GUI-KV achieves the best performance, confirming the synergistic nature of the two components. All variants of our method consistently outperform the SnapKV baseline across all screenshot counts, demonstrating the overall effectiveness of our approach. Notably, with just a 20% KV cache budget at the 7-screenshot setting, GUI-KV matches the performance of the full-cache counterpart, highlighting its efficiency. Performance for all methods peaks at 7 screenshots and declines thereafter, suggesting that excessive visual context introduces distraction. We observe that GUI-KV exhibits a steeper performance drop beyond 7 screenshots compared to full-cache, indicating that aggressive compression (20% budget) may amplify sensitivity to noisy visual context. This represents a limitation when operating under both extreme compression and excessive screenshot counts simultaneously. However, at practical operating points with moderate compression ratios (40-80% budgets as shown in Table 1), GUI-KV can effectively reduce such long-context distraction, even slightly exceeding full-cache performance on datasets like ScreenSpotV2 and AgentNetBench. This confirms that full-cache performance is actively harmed by attending to irrelevant past screenshots; GUI-KV mitigates this by discarding redundant spatial tokens, thereby cleaning the context window and occasionally surpassing the full-cache baseline. Further robustness checks, including hyperparameter sensitivity for $r$ and alternative spatial saliency mechanisms, are detailed in §C.

## 6 Conclusion

This paper studies KV cache compression for GUI agents and introduces GUI-KV, a plug-and-play approach that exploits the spatio-temporal structure of GUI interactions. Our analysis reveals that attention patterns on GUI screenshots are highly sparse and relatively uniform across layers, making heuristic layer-wise schedules designed for natural images suboptimal. Building on these insights, GUI-KV combines residual-stream saliency to better surface semantically important visual tokens with a redundancy-aware temporal scoring rule that preserves information not already captured by the most recent frame. Across six benchmarks spanning visual grounding, offline action prediction, and online evaluation, GUI-KV recovers near–full-cache accuracy at modest budgets and consistently outperforms strong baselines over a wide range of compression ratios. On UI-TARS-1.5-7B, 10–20% of the cache typically closes most of the performance gap on visual grounding, and competitive step accuracy is maintained on action-prediction tasks. Moreover, decoding compute drops substantially with negligible prefill overhead, drastically reducing the memory footprint and bandwidth requirements, enabling the deployment of GUI agents on standard consumer hardware.

## 7 Ethics statement

This work studies inference-time KV cache compression for GUI agents and adheres to the ICLR Code of Ethics. The research does not involve new data collection from human subjects, user studies, or crowdsourcing. All experiments are conducted on publicly available benchmarks for GUI agents and web automation, including ScreenSpotV2 and ScreenSpot-Pro for GUI grounding (Wu et al., 2025; Li et al., 2025), AndroidControl and Multimodal-Mind2Web for offline action prediction (Li et al., 2024a; Deng et al., 2023), AgentNetBench (Wang et al., 2025), and OSWorld-Verified for online evaluation (Xie et al., 2024). We used these datasets strictly under their respective licenses and terms of use and did not modify or augment them with additional data containing personally identifiable information. To our knowledge, the benchmarks do not include sensitive personal data; our study focuses on efficiency of inference-time caching and does not attempt to extract, infer, or reconstruct private information.

For model evaluation, we rely on existing GUI agents, UI-TARS-1.5-7B (Qin et al., 2025) and OpenCUA-7B (Wang et al., 2025), and do not introduce additional training or fine-tuning on user data. For the online setting (OSWorld-Verified), we followed the benchmark's official evaluation protocol, confined tasks to benign workflows, and did not attempt to circumvent security controls, access private accounts, or perform actions beyond the scope of the benchmark. Our method improves computational efficiency and memory usage for GUI agents; like other advances in automation, such capabilities could be misused if deployed without adequate safeguards. We encourage responsible use aligned with the Code of Ethics, organizational policies, and applicable laws, and we discourage applications that violate privacy, fairness, or safety norms.

We disclose the use of large language models only for polishing the writing of this paper (see Appendix), not for conducting experiments or generating data or labels.

## 8 Reproducibility statement

We aim to make the study reproducible by specifying models, datasets, metrics, and implementation details in the main text and appendix. The proposed method is defined in §4, with algorithmic details provided in §A (Algorithm 1). Experimental settings, including benchmarks, evaluation metrics, compared baselines, and model backbones (UI-TARS-1.5-7B and OpenCUA-7B), are described in §5.1. Hyperparameters used across experiments (e.g., rank $r$, temperature $\tau$, and trade-off coefficients) are reported in §C. Additional ablations and profiling results, including breakdown of prefill-time overhead and sensitivity to the rank parameter, are included in the appendix tables referenced therein. All datasets we evaluate on are public and cited in the paper. Together, these details are intended to enable independent verification of the reported results. To further facilitate reproducibility, our implementation will be made publicly available as open-source code upon publication.

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

# A Details of GUI-KV

We provide a detailed algorithmic description of our KV cache compression method in Algorithm 1. The algorithm combines spatial saliency guidance with temporal redundancy scoring to selectively retain the most informative tokens in the KV cache. The process consists of three main steps: (1) computing spatial saliency scores that combine attention weights with hidden state norms, (2) identifying temporally redundant tokens through subspace projection, and (3) combining both signals to select the final set of tokens for retention.

---

**Algorithm 1** GUI-KV: Spatio-Temporal KV Cache Compression for GUI Agents

---

**Require:** Full context token indices $\mathcal{C}$, partitioned into current visual $\mathcal{I}_t$, past visual $\mathcal{I}_{<t}$, and text $\mathcal{T}$.
**Require:** Per-head key matrices $\{\boldsymbol{K}^h\}_{h=1}^H$, hidden states $\boldsymbol{X}$, per-head attention scores $\{\boldsymbol{A}^h\}_{h=1}^H$.
**Require:** Hyperparameters: saliency weight $\alpha$, budget ratio $\gamma$, QR rank $r$.

1: Let $n \leftarrow |\mathcal{C}|$ be the total number of tokens.
2: Initialize sets of kept indices $\mathcal{C}_{\text{kept}}^h \leftarrow \emptyset$ for each head $h$.
3: **for** each head $h = 1, \ldots, H$ **do**
4:                  ▷ **§4.1: Compute Spatial Saliency Scores**
5:   Initialize spatial scores $\boldsymbol{\psi}^h \in \mathbb{R}^n$.
6:   **for** each token index $i \in \mathcal{C}$ **do**
7:    **if** $i \in \mathcal{I}_t$ **then**               ▷ Visual tokens
8:     $S_i \leftarrow \|x_i\|_2$         ▷ L2 norm of residual-stream hidden state
9:     $\psi_i^h \leftarrow A_i^h + \alpha S_i$
10:    **else**                  ▷ Text tokens
11:     $\psi_i^h \leftarrow A_i^h$
12:    **end if**
13:   **end for**
14:                ▷ **§4.2: Compute Temporal Redundancy Scores**
15:   Let $\boldsymbol{K}_{\mathcal{I}_t}^h$ be the key matrix rows for indices in the current frame $\mathcal{I}_t$.
16:   $(\boldsymbol{K}_{\mathcal{I}_t}^h)^\top \approx \boldsymbol{Q}_t^h \boldsymbol{R}_t^h$          ▷ Rank-$r$ QR decomposition
17:   $\boldsymbol{P}_t^h \leftarrow \boldsymbol{Q}_t^h (\boldsymbol{Q}_t^h)^\top$       ▷ Projector onto current frame's key subspace
18:   Initialize redundancy scores $\boldsymbol{\rho}^h$ for tokens in $\mathcal{I}_{<t}$.
19:   **for** each past visual token $i \in \mathcal{I}_{<t}$ **do**
20:    $k_i^h \leftarrow i$-th row of $\boldsymbol{K}^h$
21:    $\rho_i^h \leftarrow \|k_i^h(\boldsymbol{I} - \boldsymbol{P}_t^h)\|_2$      ▷ Residual norm (non-redundancy)
22:   **end for**
23:   $\tau^h \leftarrow \text{Percentile}_{1-\gamma}\big(\{\rho_j^h : j \in \mathcal{I}_{<t}\}\big)$     ▷ Redundancy threshold
24:               ▷ **Combine Scores and Select Tokens**
25:   Initialize final scores $\hat{\boldsymbol{\psi}}^h \in \mathbb{R}^n$.
26:   **for** each token index $i \in \mathcal{C}$ **do**
27:    **if** $i \in \mathcal{I}_{<t}$ **then**      ▷ Apply temporal filter to past visual tokens
28:     $\hat{\psi}_i^h \leftarrow \psi_i^h \cdot \mathbb{1}[\rho_i^h \geq \tau^h]$
29:    **else**     ▷ Keep spatial score for current visual and all text tokens
30:     $\hat{\psi}_i^h \leftarrow \psi_i^h$
31:    **end if**
32:   **end for**
33:   Let $k \leftarrow \lceil n \cdot \gamma \rceil$       ▷ Number of tokens to keep based on budget
34:   $\mathcal{C}_{\hat{\psi}^h}^h \leftarrow$ indices of top-$k$ tokens from $\mathcal{C}$ based on scores $\hat{\boldsymbol{\psi}}^h$.
35: **end for**
36: **return** $\{\mathcal{C}_{\hat{\psi}^h}^h\}_{h=1}^H$       ▷ Return the set of indices to keep for each head

---

Table 3: Compression breakdown in GFLOPs (averaged across a single instance).

| Component | GFLOPs |
|-----------|--------|
| Attention Scoring | 4.607 |
| QR Decomposition | 0.013 |
| Projection | 0.917 |
| Top-K Selection | 0.007 |
| Memory Gathering | 0.057 |

Table 4: Performance of UI-TARS-1.5-7B with GUI-KV on AgentNetBench under various budgets and rank $r$.

| Rank $r$ | Budget | | | | | | | |
|----------|--------|------|------|------|------|------|------|------|
| | 1% | 3% | 5% | 10% | 15% | 20% | 40% | 80% |
| 8 | **0.2** | 0.9 | **1.9** | 6.3 | 9.9 | 14.6 | 19.2 | 21.7 |
| 16 | **0.2** | 0.5 | 1.8 | **6.8** | 10.9 | 14.3 | 18.4 | 19.3 |
| 32 | **0.2** | **1.0** | 1.8 | 6.1 | **11.9** | 16.3 | **21.6** | **22.6** |
| 64 | **0.2** | 0.7 | 1.8 | 6.4 | 11.5 | **16.6** | 19.6 | 20.1 |
| 128 | **0.2** | 0.9 | **1.9** | 6.3 | 9.9 | 14.6 | 19.2 | 21.7 |

## B  Further Discussions

### B.1  Compression Overhead

We profile the computation introduced by GUI-KV during the prefill phase and observe that it is negligible relative to the attention scoring computation. We measure the overhead using UI-TARS-1.5-7B on the AgentNetBench benchmark. As summarized in Table 3, attention scoring dominates the cost, while QR decomposition, top-$k$ selection (0.007 GFLOPs; 0.15%), and memory gathering (0.057 GFLOPs; 1.22%) together contribute under 2% overhead. Consequently, the QR-based subspace projection in our proposed method adds a vanishingly small fraction to prefill-time computation and does not hinder throughput.

### B.2  The Impact of $r$ in Temporal Redundancy Scoring

We aim to understand the impact of different values of $r$. We vary different values of $r$ and test UI-TARS-1.5-7B on AgentNetBench. Table 4 show the results. We found that various values of $r$ does not yield significantly different results as all settings outperforming the other baselines in Table 1. Among these values, $r = 32$ results in the best overall performance.

### B.3  Effectiveness of Other Spatial Saliency Guidance

In the early stage of our study, we experimented with other spatial saliency guidance. Essenatially, we can replace $S_i$ discussed in §4.1 with other methods that indicate spatial saliency. This includes: (1) **Pixel Histogram Entropy**: we transform the current screenshot into gray-scale image. Then, we compute histogram probability for each color bin $b, b \in \{1, ..., 256\}$. $S_i$ for this method is defined as computing the entropy of each histogram probability for each image patch. (2) **Sobel Filter**: Sobel Filter (Sobel, 1968) is a classic operator for detecting edges in an image. $S_i$ is defined as the output of a $3 \times 3$ Sobel operator. (3) **Center-Surround Contrast**: For each patch, we compare its average color to the average color of a larger surrounding region. A large difference implies high saliency. Concretely, we convert the image from RGB to CIELAB, in which the Euclidean distance between two colors approximates the perceived difference to the human eye.

The results are summarized in Table 5. We observe that different guidance approaches have their own strengths. For example, Center-Surround Contrast is most effective in the 10-20%, while Pixel Histogram

Table 5: Performance of UI-TARS-1.5-7B with GUI-KV on ScreenSpot-V2 using different spatial saliency guidance under various budgets.

| Spatial Saliency Guidance | Budget | | | | | | | |
|---|---|---|---|---|---|---|---|---|
| | 1% | 3% | 5% | 10% | 15% | 20% | 40% | 80% |
| Pixel Histogram Entropy | **17.3** | 61.1 | 76.2 | 83.5 | 85.5 | 86.8 | 88.1 | 88.9 |
| Sobel Filter | 1.2 | 10.3 | 21.5 | 33.7 | 43.2 | 46.4 | 80.1 | 87.8 |
| Center-Surround Contrast | 17.0 | 60.1 | 76.7 | **85.6** | **86.1** | **88.0** | 87.7 | 88.3 |
| Residual Stream L2 Norm (§4.1) | 16.7 | **63.6** | **79.1** | 83.5 | 85.7 | 87.9 | **88.2** | **89.3** |

Entropy performs the best at 1% budget. Overall, the best-performing method is the L2 Norm of the residual stream outlined in §4.1.

## C  Hyper-parameters Selection

In this section, we illustrate the values of the hyper-parameters used in our experiments. We set the QR rank to $r = 32$ so that the projector in §4.2 spans a compact representation of the latest frame. The temperature for normalizing residual-stream norms in Equation (4) is fixed at $\tau = 3.5$, and the saliency-to-attention trade-off in the same scoring rule uses $\alpha = 2$ (Equation (5)). Finally, we retain $\omega = 8$ observation tokens when forming the queries that drive token scoring, as described in §3.

## D  Large Language Models Usage Statement

Large language models are only used for polishing the content of this paper.

