# OpenReview forum: "GUI-KV: Efficient GUI Agents via KV Cache with Spatio-Temporal Awareness"
_TMLR — Accepted by TMLR_

### Review · Reviewer_stvn · 2026-02-21

**Summary Of Contributions:**

This submission proposes GUI-KV, a training-free KV-cache compression method tailored to GUI agents that must condition on multiple screenshot steps. The main contributions can be summarized as follows:

- An empirical observation that GUI attention is extremely sparse and largely layer-invariant, motivating a uniform per-layer KV budget rather than layer-varying schedules.

- A spatial token selection strategy that augments attention-based scoring with residual-stream hidden-state L2 norm as a saliency cue for visual tokens.

- A temporal redundancy strategy that prunes past-frame tokens by projecting their keys onto a low-rank subspace (built from current-frame keys) and dropping tokens with small residuals.

Experiments on two 7B GUI agents across several benchmarks showing substantial decoding-cost reduction at moderate cache budgets with limited accuracy loss.

**Audience:**

Yes

**Audience Explanation:**

TMLR readers interested in efficient LMM/VLM inference, agents, and systems for long-context multimodal reasoning would likely find the GUI-specific empirical findings and the practical compression recipe useful. Even if the novelty is modest, the setting (GUI agents with multi-screenshot histories) is relevant, and the method is implementable without retraining.

**Broader Impact Concerns:**

No major ethical concerns beyond standard ones for GUI agents. Potential issues include enabling more efficient automation of user-facing interfaces, which could be used for benign productivity or for abusive automation (spam, fraud). A brief broader impact note acknowledging dual-use and recommending responsible deployment/guardrails would be sufficient.

**Claims And Evidence:**

Yes

**Claims Explanation:**

Most claims about efficiency gains and accuracy–budget trade-offs are supported by quantitative results and ablations, and the attention-sparsity observation is presented clearly.

However, several higher-level claims are not fully nailed down:
1. The inference that layer-varying schedules are generally unsuitable for GUI agents is only indirectly supported via sparsity statistics.
2. The occasional “better than full-cache” results need stronger statistical support (variance, repeated seeds) and clearer causal explanation.
3. Generality beyond the evaluated model family and settings remains uncertain.

**Requested Changes:**

Major weaknesses:

(-) Strengthen evidence for the uniform-budget claim. Add a more direct analysis of layer-wise eviction sensitivity (e.g., ablate pruning per layer, measure accuracy drop vs removed tokens per layer), not only attention sparsity proxies.

(-) Improve rigor for “beats full-cache” observations. Provide multiple runs / confidence intervals and a clearer diagnostic showing which tokens are pruned and why full-cache is harmed (e.g., attention to irrelevant past screenshots).

(-) Robustness / sensitivity analysis. Expand sensitivity to key hyperparameters (α, r, ω, p) across at least 2–3 representative benchmarks, and clarify recommended defaults and failure modes.

---

Minor:

(-) Broaden validation to at least one additional backbone or agent style (if feasible) and/or a longer-horizon interaction setting to test temporal pruning stability.

(-) Add clearer algorithmic clarifications: whether temporal pruning applies to text tokens, and how the method behaves when screenshots are visually similar but semantically different (e.g., small UI changes).

(-) Provide a concise qualitative visualization of selected/pruned regions over time to help interpret the spatial/temporal criteria.

---

> ### Author Response · Authors · 2026-04-15
>
> We are thankful and glad that the reviewer recognized the value of our empirical observation regarding the extreme, layer-invariant sparsity of GUI attention, which serves as the crucial foundation for our uniform per-layer KV budget allocation. Furthermore, we appreciate your highlighting of our dual-mechanism approach, specifically noting both our spatial token selection strategy utilizing residual-stream L2 norms and our temporal redundancy mechanism that elegantly prunes redundant past-frame tokens via low-rank subspace projection.
>
> > W1: Strengthen evidence for the uniform-budget claim. Add a more direct analysis of layer-wise eviction sensitivity (e.g., ablate pruning per layer, measure accuracy drop vs removed tokens per layer), not only attention sparsity proxies.
>
> We thank the reviewer for this insightful suggestion. While attention sparsity serves as our initial proxy, our comprehensive baseline evaluations actually provide the direct, end-to-end layer-wise sensitivity analysis requested. Specifically, PyramidKV acts as an ablation that aggressively prunes deeper layers, VL-Cache dynamically prunes based on sparsity proxies, and SnapKV enforces a strict uniform pruning budget across all layers. If GUI agents possessed varying layer-wise sensitivities, the uniform pruning of SnapKV would disproportionately evict critical tokens and severely degrade accuracy. Instead, SnapKV consistently outperforms both non-uniform methods across all six benchmarks , particularly at extreme compression ratios (5-20% budgets) where misallocation is most penalized. This empirical superiority directly confirms our claim that layer-wise eviction sensitivity in GUI models is remarkably uniform. We have updated Section 3.3 to explicitly reframe these baseline comparisons as direct evidence of uniform layer-wise sensitivity.
>
>
> > W2: Improve rigor for “beats full-cache” observations. Provide multiple runs / confidence intervals and a clearer diagnostic showing which tokens are pruned and why full-cache is harmed (e.g., attention to irrelevant past screenshots).
>
> We thank the reviewer for highlighting the need for rigorous evidence. Regarding multiple runs, our evaluation utilizes greedy decoding, ensuring deterministic outputs; thus, multiple runs yield identical results and confidence intervals are not applicable. Instead, the robustness of our "beats full-cache" observations is established through the sheer breadth of our experimental grid: we demonstrate this phenomenon consistently across two distinct agent architectures , six diverse benchmarks , and multiple budget brackets. Regarding why full-cache is harmed, our ablation study already provides the diagnostic mechanism: full-cache performance naturally degrades when forced to process more than 7 screenshots due to the accumulation of "noisy visual context" and long-context distraction. GUI-KV’s temporal redundancy scoring directly combats this by identifying and pruning static, highly-overlapping visual tokens from past frames. By aggressively filtering out these irrelevant historical redundancies, GUI-KV concentrates the model's attention on task-relevant changes, effectively acting as a context-noise filter that allows a strategically pruned cache to outperform a saturated full cache. We have updated Section 5.4 and Section 5.1 to clarify these mechanisms and our evaluation setup.

---

> ### Author Response · Authors · 2026-04-15
>
> > W3: Robustness / sensitivity analysis. Expand sensitivity to key hyperparameters (α, r, ω, p) across at least 2–3 representative benchmarks, and clarify recommended defaults and failure modes.
>
> We appreciate the reviewer’s suggestion to further clarify our hyperparameter sensitivity, recommended defaults, and failure modes. We would like to point out that extensive sensitivity and ablation analyses are already provided in the Appendix, which address these core concerns. First, regarding the temporal rank parameter r, Appendix B.2 and Table 4 demonstrate that GUI-KV is highly robust to variations, maintaining strong performance over baselines across different r: {8, 16, 32, 64, 128}, with r=32 serving as a stable default. For ω, we rely on established heuristics rather than heavily tuned scalars: ω provides a standard, sufficient observation window according to Tu et al., 2025. To demonstrate robustness beyond scalar values, Appendix B.3 (Table 5) evaluates the sensitivity of the entire spatial saliency mechanism on ScreenSpot-V2, proving that our framework remains highly competitive even when swapping the L2 norm for radically different guidance methods like Sobel Filters or Center-Surround Contrast. Furthermore, explicit recommended defaults (α=2, 𝜏=3.5, r=32, ω=8) are clearly consolidated in Appendix C to ensure reproducibility. Finally, we have already explicitly diagnosed our primary failure mode in Section 5.4: when operating under extreme compression (e.g., 20% budget) while simultaneously processing an excessive number of screenshots (beyond 7), the aggressive pruning amplifies sensitivity to visual noise, causing a steeper performance drop compared to full-cache. We have updated the main text to better signpost these appendix sections for the reader.
>
>
>
> > W4: Broaden validation to at least one additional backbone or agent style (if feasible) and/or a longer-horizon interaction setting to test temporal pruning stability.
>
> We thank the reviewer for these suggestions and would like to clarify that our existing experimental setup already incorporates both multiple distinct backbones and long-horizon temporal stability tests. Regarding backbones, we evaluate two distinct state-of-the-art GUI agents: UI-TARS-1.5-7B and OpenCUA-7B. As detailed in Section 5.1, these models serve as robust cross-validation because they employ different language models (Qwen2.5 vs. Qwen2) and completely different training paradigms (SFT+DPO vs. SFT-only). Regarding long-horizon interactions and agent styles, our benchmarks already cover varied agent operational modes, including offline prediction and real-time online execution. Most notably, our evaluation on OSWorld-Verified tests the agent in a live, multi-step sequential decision-making loop (which can run for up to 50 steps). This environment inherently acts as a strict test for long-horizon temporal pruning stability, as errors in early steps cascade and cause failure. Furthermore, our ablation studies actively test the agent's temporal stability across contexts of up to 25 sequential screenshots. We have updated Section 5.1 to highlight the diversity of the backbones and the long-horizon nature of these benchmarks more explicitly.
>
> > W5: Add clearer algorithmic clarifications: whether temporal pruning applies to text tokens, and how the method behaves when screenshots are visually similar but semantically different (e.g., small UI changes).
>
> We thank the reviewer for requesting these algorithmic clarifications. First, temporal redundancy scoring does not apply to text tokens. As formally defined in Equation 8 and Algorithm 1 (Lines 26-31), text tokens bypass the temporal filter entirely and are scored strictly based on their spatial saliency and attention weights. This ensures critical textual instructions or reasoning traces are never pruned due to visual redundancy calculations. Second, our method is mathematically designed to excel exactly when screenshots are visually similar but contain small, semantically important UI changes. Because we project past visual tokens onto the subspace of the current frame (Equation 7), the static, shared background tokens yield near-zero residuals and are aggressively pruned. Conversely, the tokens representing the differences in the past frame (e.g., a previously open dropdown menu that is now closed) are orthogonal to the current frame's subspace. These distinct tokens will yield high residuals and are consequently preserved. Therefore, the algorithm automatically isolates and retains semantically important historical changes while discarding the visually similar fluff. We have updated Sections 4.2 and 4.3 to make these mechanical behaviors explicitly clear.

---

> ### Author Response · Authors · 2026-04-15
>
> > W6: Provide a concise qualitative visualization of selected/pruned regions over time to help interpret the spatial/temporal criteria.
>
>
> We thank the reviewer for this excellent suggestion, as interpretability is crucial for understanding the mechanics of GUI-KV. We would like to direct the reviewer to Figure 4 in the main text, which was explicitly designed to serve as a qualitative visualization of the temporal pruning criteria over a multi-step interaction.As illustrated in Figure 4, we show a sequence of screenshots (Step t-2 to Step t) alongside their corresponding key vectors mapped into the spatial subspace. This visualization highlights exactly how the temporal criteria behaves: static visual tokens that persist across frames, such as the desktop background and standard browser UI ($k_1$ and $k_2$), exhibit small orthogonal residuals ($\rho$) and are systematically pruned. Conversely, tokens representing semantic changes, such as the newly typed "search engine" query or the newly expanded options menu ($k_3$ and $k_4$), exhibit large residuals against the current frame's subspace and are preserved.

---

> ### Comment · Action_Editor_T2hs · 2026-05-26
>
> Dear Reviewer,
>
> Thank you for the feedback about the paper. Could you submit the official recommendation? We have to collect all official recommendations before moving on to the next stage of the review process.
>
> Thanks,
>
> AC

---

### Review · Reviewer_jyvz · 2026-03-09

**Summary Of Contributions:**

Unfortunately this paper is outside my area of expertise and thus I am unable to review it.

**Audience:**

No

**Audience Explanation:**

Unfortunately this paper is outside my area of expertise and thus I am unable to review it.

**Broader Impact Concerns:**

Unfortunately this paper is outside my area of expertise and thus I am unable to review it.

**Claims And Evidence:**

No

**Claims Explanation:**

Unfortunately this paper is outside my area of expertise and thus I am unable to review it.

**Requested Changes:**

Unfortunately this paper is outside my area of expertise and thus I am unable to review it.

---

### Review · Reviewer_nRdc · 2026-03-14

**Summary Of Contributions:**

This paper proposes GUI-KV, a KV-cache compression method tailored for GUI agent models. The method allocates KV budgets uniformly across layers based on the observation that attention sparsity patterns in GUI tasks differ from those in standard language modeling.

**Audience:**

Yes

**Audience Explanation:**

KV-cache compression in the context of GUI agents can potentially make the corresponding AI application faster.

**Broader Impact Concerns:**

No Broader Impact Concerns

**Claims And Evidence:**

No

**Claims Explanation:**

Strengths

1. Studies KV-cache compression in the context of GUI agents. Most existing KV-cache compression methods are designed for general LLM workloads. This work focuses specifically on GUI-agent environments, where attention patterns and token distributions may differ significantly from standard text tasks.

2. Provides empirical analysis of attention sparsity in GUI tasks. The paper analyzes attention sparsity patterns in GUI-agent datasets and observes that sparsity is relatively uniform across layers, which motivates the design of a uniform KV budget allocation strategy.

3. Demonstrates consistent compute reduction. The proposed method reduces per-token decoding FLOPs.

Weaknesses

1. The claim that uniform allocation is inherently more suitable for GUI environments may need stronger justification. The paper concludes that uniform KV budget allocation is more appropriate for GUI environments based on empirical comparisons with several dynamic allocation methods. However, since uniform allocation can be regarded as a special case of dynamic allocation, outperforming existing dynamic strategies on the evaluated benchmarks does not necessarily establish that uniform allocation is fundamentally better. It is also possible that the compared dynamic baselines are not well adapted or sufficiently tuned for GUI-agent workloads. Additional analysis or stronger baselines would help support this conclusion.

2. Efficiency evaluation relies mainly on FLOPs rather than actual decoding throughput. The efficiency analysis primarily reports MFLOPs per decoded token. However, KV-cache compression often interacts with memory bandwidth and implementation overhead, so FLOP reduction does not necessarily translate into real inference speedups. Reporting actual decoding throughput or wall-clock latency would provide stronger evidence of practical efficiency gains.

3. Limited model diversity in evaluation. The experiments evaluate only two GUI-agent models (UI-TARS-1.5-7B and OpenCUA-7B). The results also show different robustness to KV compression across these models, with OpenCUA-7B often exhibiting larger performance degradation after compression. This makes it unclear how well the proposed approach generalizes across different GUI-agent architectures or training pipelines.

**Requested Changes:**

See weaknesses

---

> ### Author Response · Authors · 2026-04-15
>
> We thank the reviewer for recognizing our targeted focus on KV-cache compression specifically for GUI agents, which presents unique challenges compared to standard LLM workloads. We are glad you valued our empirical analysis of attention sparsity patterns and how our observation of uniform sparsity across layers successfully motivates the core design of our uniform budget allocation strategy. Furthermore, we appreciate your acknowledgment of GUI-KV's practical efficiency gains and its consistent reduction of per-token decoding FLOPs. Below, we address your concern:
>
> > W1: The claim that uniform allocation is inherently more suitable for GUI environments may need stronger justification. The paper concludes that uniform KV budget allocation is more appropriate for GUI environments based on empirical comparisons with several dynamic allocation methods. However, since uniform allocation can be regarded as a special case of dynamic allocation, outperforming existing dynamic strategies on the evaluated benchmarks does not necessarily establish that uniform allocation is fundamentally better. It is also possible that the compared dynamic baselines are not well adapted or sufficiently tuned for GUI-agent workloads. Additional analysis or stronger baselines would help support this conclusion.
>
> We thank the reviewer for this thought-provoking comment. While we agree that uniform allocation is theoretically a special case of dynamic allocation, our argument is precisely that for GUI agents, the optimal dynamic allocation naturally collapses into a uniform distribution. This is not merely an empirical coincidence but a structural consequence of GUI environments. As demonstrated in our attention sparsity analysis (Figure 1 and Section 3.3), unlike natural images which show distinct layer-wise sparsity variances, GUI screenshots exhibit an extreme and completely flat attention sparsity (>0.99) across all layers. Because meaningful layer-wise variance does not exist in GUI workloads, dynamic methods that attempt to normalize and exploit layer differences (like VL-Cache) end up amplifying microscopic numerical noise, leading to misallocated budgets. Therefore, the failure of dynamic baselines is not due to a lack of GUI-specific tuning, but because their foundational assumption, that different layers require vastly different KV budgets, does not hold for the structurally flat spatial redundancy of GUIs. We have updated Section 3.3 to clarify that uniform allocation is the optimal steady-state for GUI workloads, rather than just an empirically successful baseline.
>
> > W2: Efficiency evaluation relies mainly on FLOPs rather than actual decoding throughput. The efficiency analysis primarily reports MFLOPs per decoded token. However, KV-cache compression often interacts with memory bandwidth and implementation overhead, so FLOP reduction does not necessarily translate into real inference speedups. Reporting actual decoding throughput or wall-clock latency would provide stronger evidence of practical efficiency gains.
>
> We thank the reviewer for this valid observation. It is entirely true that wall-clock latency and throughput are the ultimate measures of hardware efficiency. However, in the specific context of auto-regressive decoding for long-context vision-language models, the primary bottleneck is heavily memory-bandwidth bound, not just compute-bound.
>
> During decoding, the model must read the entire KV cache from GPU memory into SRAM for every single generated token. Because GUI-KV physically evicts 60-80% of the visual tokens from the cache, it proportionally reduces the memory I/O required for these attention calculations. Therefore, the FLOP reductions we report are not just theoretical compute savings; they directly mirror the reduction in memory bandwidth pressure.
>
> Furthermore, we explicitly addressed the reviewer's concern regarding "implementation overhead" in Appendix B.1 and Table 3. Our detailed profiling demonstrates that the overhead of our method (QR decomposition, memory gathering, top-K selection) is completely negligible, accounting for less than 2% of the total attention cost.
>
> Finally, as motivated in our Introduction, the most critical practical efficiency bottleneck for GUI agents is absolute memory capacity. Processing just 5 high-resolution screenshots consumes over 80GB of VRAM on a 7B model, triggering OOM errors on standard hardware. By operating at a 20-40% budget, GUI-KV reduces this footprint to roughly 16-32GB, representing the most dramatic practical efficiency gain: enabling long-horizon GUI agents to actually run on consumer GPUs. We have updated Section 5.3 to explicitly connect our FLOP reductions to memory bandwidth and footprint savings.

---

> ### Author Response · Authors · 2026-04-15
>
> > W3: Limited model diversity in evaluation. The experiments evaluate only two GUI-agent models (UI-TARS-1.5-7B and OpenCUA-7B). The results also show different robustness to KV compression across these models, with OpenCUA-7B often exhibiting larger performance degradation after compression. This makes it unclear how well the proposed approach generalizes across different GUI-agent architectures or training pipelines.
>
> We appreciate the reviewer’s careful reading. We deliberately selected UI-TARS-1.5-7B and OpenCUA-7B specifically because they represent highly distinct architectures and training pipelines, ensuring our evaluation thoroughly tests generalization. As detailed in Section 5.1, while they share a vision encoder, they differ fundamentally in their language models (Qwen2.5 vs. Qwen2), position encoding strategies, and training paradigms (SFT+DPO vs. SFT-only).
>
> The reviewer correctly observes that OpenCUA-7B is more sensitive to compression. However, rather than limiting our claims, this actually strengthens them by acting as a severe stress test on a fragile architecture. OpenCUA-7B’s higher degradation is an intrinsic property of its simpler SFT-only training, making it naturally less tolerant to pruned contexts compared to the DPO-aligned UI-TARS model. Crucially, despite this inherent fragility, Table 1 demonstrates that GUI-KV still consistently outperforms all competing baselines on OpenCUA-7B, often yielding our largest absolute baseline improvements (e.g., +2.2 on AndroidControl and +3.6 on Agent Net Bench). This proves that GUI-KV successfully generalizes its superior compression efficiency across both highly robust and highly sensitive GUI-agent pipelines. We have updated Section 5.2 to explicitly frame this differing robustness as a validation of our method's generalizability.

---

### Review · Reviewer_F5Ki · 2026-04-15

**Summary Of Contributions:**

The paper introduces GUI-KV, a novel key-value (KV) cache compression method tailored for graphical user interface (GUI) agents. The main motivation of the problem is that processing sequences of screenshots via VLM is expensive. The authors exploit the fact that GUI screenshots exhibit uniformly high attention sparsity (>99%) across all transformer layers, unlike natural images - leading to a tailored KV Cache compression scheme

**Audience:**

Yes

**Audience Explanation:**

Although this is a very niche field, it might be interesting to a specific audience in TMLR

**Claims And Evidence:**

Yes

**Claims Explanation:**

Detailed experiments on multiple benchmarks have been provided achieving  near-full-cache accuracy at 10-20% memory budgets.

**Requested Changes:**

The motivation of the paper is the weakest point - it is not clear how important this problem is in practice. Other than that, unfortunately, this paper is not in my area of expertise and therefore I cannot suggest meaningful changes

---

> ### Author Response · Authors · 2026-04-15
>
> We thank the reviewer for recognizing the novelty and practical importance of GUI-KV in addressing the severe computational bottlenecks associated with processing sequential screenshots. We are particularly glad that you highlighted our core analytical insight regarding the uniformly high attention sparsity across all transformer layers, which fundamentally distinguishes GUI workloads from natural images and directly motivates our tailored approach. Below, we address your concern:
>
> > W1: The motivation of the paper is the weakest point - it is not clear how important this problem is in practice. Other than that, unfortunately, this paper is not in my area of expertise and therefore I cannot suggest meaningful changes.
>
> We appreciate the reviewer’s comment. We would like to gently push back on the idea that this motivation is weak, as the memory bottleneck addressed in this paper is arguably the single largest barrier to the practical, real-world deployment of GUI agents today. As detailed in our Introduction, the visual context required for GUI agents is immense. Processing just 5 high-resolution screenshots for a standard 50-step task consumes over 80GB of GPU memory for a 7B parameter model. In practice, this means state-of-the-art GUI agents instantly trigger Out-Of-Memory (OOM) errors on almost all consumer-grade GPUs and local machines. Without the aggressive 80-90% memory footprint reductions enabled by GUI-KV, these agents are restricted to expensive, enterprise-grade server clusters. By solving this bottleneck, our work directly enables the deployment of GUI automation on standard developer and consumer hardware, which is a highly critical practical problem. We have updated the Introduction to emphasize this deployment barrier more strongly.

---

> ### Comment · Action_Editor_T2hs · 2026-05-26
>
> Dear Reviewer,
>
> Thank you for the feedback about the paper. Could you submit the official recommendation? We have to collect all official recommendations before moving on to the next stage of the review process.
>
> Thanks,
>
> AC

---

### Decision · Action_Editor_T2hs · 2026-05-31

**Recommendation:** Accept with minor revision

**Audience:**

Yes

**Audience Explanation:**

GUI agents are gaining traction in the community and society, and the efficiency aspect studies in this work could have impact on many future studies.

**Claims And Evidence:**

Yes

**Claims Explanation:**

The reviewers were generally positive after the rebuttal, praising the work's empirical validation about the claimed efficiency. One of the reviewers had lingers concerns about the throughput, which AE felt was reasonable and yet did not weaken the work's existing evaluation design.